# Consensus Label Propagation with Graph Convolutional Networks for Single-Cell RNA Sequencing Cell Type Annotation

**Daniel P Lewinsohn**
Oregon Health and Science University
Colorado College
lewinsda@ohsu.edu

**Donald F Conrad**
Oregon Health and Science University
conradon@ohsu.edu

**Cory B Scott**
Colorado College
cbs@coloradocollege.edu

## Abstract

Single-cell RNA sequencing (scRNA-seq) data, annotated by cell type, is useful in a variety of downstream biological applications, such as profiling gene expression at the single-cell level. However, manually assigning these annotations with known marker genes is both time-consuming and subjective. We present a Graph Convolutional Network (GCN) based approach to automate the annotation process. Our process builds upon existing labeling approaches, using state-of-the-art tools to find highly-confident cells through consensus and spreading these confident labels with a semi-supervised GCN. Using simulated data and two scRNA-seq data sets from different tissues, we show that our method improves accuracy over a simple consensus algorithm and the average of the underlying tools. We also demonstrate that our GCN method allows for feature interpretation, revealing important genes for cell type classification. We present our completed pipeline, written in Pytorch, as an end-to-end tool for automating and interpreting the classification of scRNA-seq data.

## 1 Introduction

Single-cell RNA sequencing (scRNA-seq) measures the RNA from each gene present in an individual cell, serving as a proxy for gene expression. High-quality labels of cell type based on the transcriptional profile produced by scRNA-seq have proven valuable for characterizing gene expression of cells, and for discovering cell types and genetic drivers of disease. Traditionally, these labels are produced by unsupervised clustering followed by labeling clusters with known marker genes. However, unsupervised clustering is limited by issues such as the size of scRNA-seq data sets as well as subjectivity in reclustering and biological interpretation of clusters [1].

The limitations of traditional cell type annotation methods have necessitated the development of automated methods for cell labeling. Three main categories of tools have emerged: marker gene based, correlation based, and supervised classification based [2]. Marker based approaches employ known marker genes for labeling, while correlation and supervised learning based approaches require manually labeled scRNA-seq data sets with the cell types of interest. Within these broad categories, the performance of individual tools varies widely across data sets [3]. As a result, using the consensus of multiple classification tools could yield higher accuracy. However, there currently exist no tools for researchers to easily apply multiple classification algorithms to their scRNA-seq data.

We address these issues in two ways. First, we provide a pipeline for annotation of scRNA-seq data with multiple state-of-the-art annotation algorithms. Second, we implement and test a semi-supervised

D. Lewinsohn et al., Consensus Label Propagation with Graph Convolutional Networks for Single-Cell RNA Sequencing Cell Type Annotation (Extended Abstract). Presented at the First Learning on Graphs Conference (LoG 2022), Virtual Event, December 9–12, 2022.

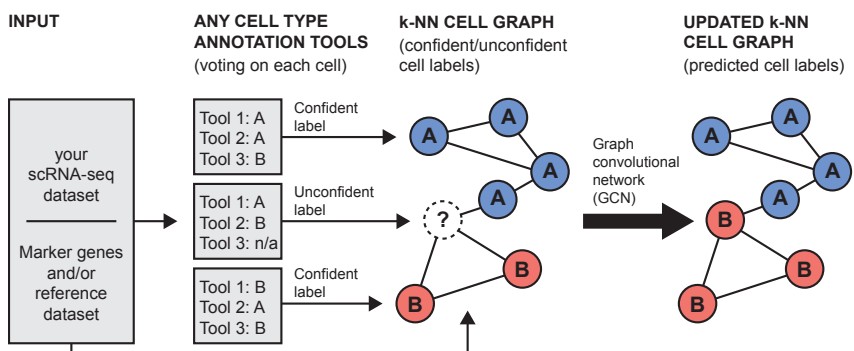

**Figure 1:** Starting from an scRNA-seq data set, a user can apply any number of tools which classify cell types (for example, classification from pre-labeled reference data or from reference genetic markers). If a majority of these underlying tools assign the same label to a cell, we say the ensemble is confident in this label. Our GCN learns to propagate labels from confidently labeled cells to the rest of the cells ("unconfident cells") via message passing in a K-nearest neighbor graph.

Graph Convolutional Network (GCN) as a mechanism to propagate labels from confidently labeled cells to unconfidently labeled cells. We show our method improves overall classification accuracy (and, more specifically, classification accuracy on unconfidently labeled cells) compared to taking the consensus of the labels from the underlying tools. We also demonstrate the use of DeepLIFT [4] as an effective interpretation tool for our GCN model, allowing researchers insight into classification decisions and important cell type gene markers.

## 2   Our Model

**Picking Confident Labels via Consensus.**   Our method first involves picking confident labels for a subset of cells in a given data set. Our pipeline currently includes five different state-of-the-art annotation methods: SCINA [5], ScType [6], ScSorter [7], SingleR [8], and ScPred [9]. These methods classify cells via clustering, specific marker genes, similarity to a reference data set, or a mix of all three - see Appendix D for an in-depth discussion of these tools. Our pipeline also allows researchers to upload their own predictions and utilize other tools. We designate a cell as being confidently labeled (and keep that cell's label) if a majority of tools agree on that label. We also compare all methods to a non-parametric label propagation approach (described in Appendix G).

**Semi-supervised GCN.**   We construct a GCN with $l$ EdgeConv [10] layers with SiLU activation function and summation aggregation. Each layer propagates embedding vectors between each node (featurized in PCA space) and its $k$ nearest neighbors (including itself). A final linear layer projects node embeddings into label space (whose dimension is the number of cell types in our data set). For architecture details see Appendix A. We train our GCN for 150 epochs, with the Adam optimizer [11] at a learning rate of $0.0001$. Our training loss is Cross-Entropy loss on the set of confidently labeled cells. Nearest neighbors are generated separately for each batch of each epoch.

**Interpretation with DeepLIFT.**   We employ DeepLIFT [4] with the Rescale rule as implemented by Captum [12]. We use the same hyperparameters batch size $b$, neighbors $k$, number of message passing steps $l$, and final embedding layer size $e$ as used during training. DeepLIFT uses the gradient of a neural network's outputs with respect to inputs to determine how much a given classification depended on a given input variable (in our case, how much classification of a given cell type depends on each gene). Calculating these attribution scores is only possible for a differentiable model, like our GCN - not possible for any of the underlying tools. We note here that the pipeline we analyzed with DeepLIFT included the PCA step, resulting in attribution scores per gene for each cell.

## 3   Data Sets

**Preprocessing Data.**   The initial input to our pipeline is a scRNA-seq count matrix $X$ where $X_{ng}$ corresponds to observed gene counts for gene $g$ in cell $n$. Cells expressing $< 200$ genes and genes

**Table 1:** Accuracy percent scores for all data sets for both all cells and "unconfident cells": cells for which the underlying methods did not have consensus. GCN accuracies are the mean $\pm$ standard deviation of accuracies from five randomly initialized trials.

| Method | Simulation 0.7 | | Simulation 0.8 | | Testis | | PBMC | |
|---|---|---|---|---|---|---|---|---|
| | **All** | **Unconf.** | **All** | **Unconf.** | **All** | **Unconf.** | **All** | **Unconf.** |
| Ours (GCN) | **90.0 $\pm$ .61** | 66.0 $\pm$ 3.9 | **96.1 $\pm$ .22** | **83.3 $\pm$ 2.6** | 86.2 $\pm$ .12 | 80.9 $\pm$ 1.8 | **93.1 $\pm$ .05** | 70.6 $\pm$ 1.5 |
| Max Consensus | 86.4 | 42.9 | 91.2 | 25.9 | 80.8 | 0.0 | 91.2 | 6.8 |
| Tool Avg. | 69.3 $\pm$ 14 | 33.1 $\pm$ 32 | 76.7 $\pm$ 12 | 34.6 $\pm$ 28 | 72.1 $\pm$ 16 | 21.7 $\pm$ 36 | 75.0 $\pm$ 6.1 | 36.9 $\pm$ 34 |
| ScType | 64.8 | 13.5 | 79.9 | 29.4 | 84.8 | 63.0 | 85.5 | **81.8** |
| ScSorter | 85.8 | 51.3 | 88.7 | 38.8 | 77.5 | 2.2 | 71.5 | 65.9 |
| SCINA | 53.7 | 7.7 | 58.7 | 2.4 | 53.9 | 0.0 | 73.3 | 6.2 |
| SingleR | 83.1 | **80.1** | 85.2 | 77.6 | $NA$ | $NA$ | 70.3 | 13.4 |
| ScPred | 59.2 | 12.8 | 71.0 | 24.7 | $NA$ | $NA$ | 74.4 | 17.1 |
| Non-Parametric | 30.6 | 16.7 | 37.4 | 15.3 | **87.4** | **84.8** | 87.9 | 70.8 |

expressed in $< 3$ cells are removed, and $X$ is row-normalized according to $x_i = log(1 + ((x_i * 10000)/\Sigma x_i)$. Both of these steps are common scRNA-seq preprocessing steps [13][14]. Finally, we use Principal Components Analysis (PCA) to project $X$ down to 500 features per cell.

**Simulated Data.** We generated our simulated data sets with Splatter [15] and parameters estimated from 4000 Pan T Cells from a healthy donor [16]. Each simulated data set contains 1000 cells, evenly split between four cell types with different transcriptomic profiles. To demonstrate that our pipeline can also incorporate reference-based tools (like SingleR and ScPred), we also generated 1000 reference cells for each data set with the same gene profiles (separated from the original data with a batch.facScale of 0.5 to simulate batch effects). Simulated data sets vary by the de.facScale parameter which determines the magnitude of variation in gene expression profile between cell type groups. Five markers were selected randomly from the top ten differentially expressed genes from each cell type. The 0.7 de.facScale and 0.8 de.facScale simulated data sets had 156 and 85 unconfidently labeled cells respectively.

**Real Data.** It is not usually feasible to acquire ground truth labels for scRNA-seq data. An alternative gold standard is Fluorescence-activated Cell Sorting (FACS), which pre-sorts cells by markers prior to conduction of scRNA-seq [17]. We test our model on two FACS-labeled data sets.

First, we use a scRNA-seq data set generated from mouse testis cells [18]. This data contains three cell types: 292 Spermatogonia, 244 Spermatocytes, and 156 Spermatids after filtering. Cell type markers were selected from relevant literature [19] and no reference data set was used for this data. There were 46 unconfidently labeled cells after prediction tool voting on the data set.

Second, we use an scRNA-seq data set generated from Human peripheral blood mononuclear cells (PBMCs) [20]. This data contains ten cell types, however, we removed cell types not purely sorted by FACS, combined CD4+ T cells, and combined CD8+ T Cells. This resulted in five cell types: 9,106 B Cells, 2,341 Monocytes, 7,572 Natural killer (NK) Cells, 38,006 CD4+ T Cells, and 19,856 CD8+ T Cells. We used the same markers as ScSorter [7] and used the 10X PBMC 3k data set as a reference as in [21]. This data set contained 2,310 unconfidently labeled cells.

## 4 Results

### 4.1 Accuracy on Test Sets

**Experiment Settings.** For the simulated and testis data sets, 20 percent of confidently labeled cells were masked and held out as a validation set. We performed a hyperparameter optimization search (see Appendix A for details) for options of batch size $b$, neighbors $k$, layers $l$, and embedding layer size $e$, selecting the GCN architecture with the highest validation accuracy. For the PBMC data set, five percent of the data set was used for this same hyperparameter optimization process as using the entire data set was computationally intractable. Each GCN used EdgeConv feature propagation between each node and its $k$ closest neighbors, with distance determined dynamically between node features (including the PCA features at the first layer). Five random initializations of the optimal model were then trained for 150 epochs as described above and mean accuracy was recorded. We use max consensus, the non-parametric neighbor majority approach, and other tool accuracies as

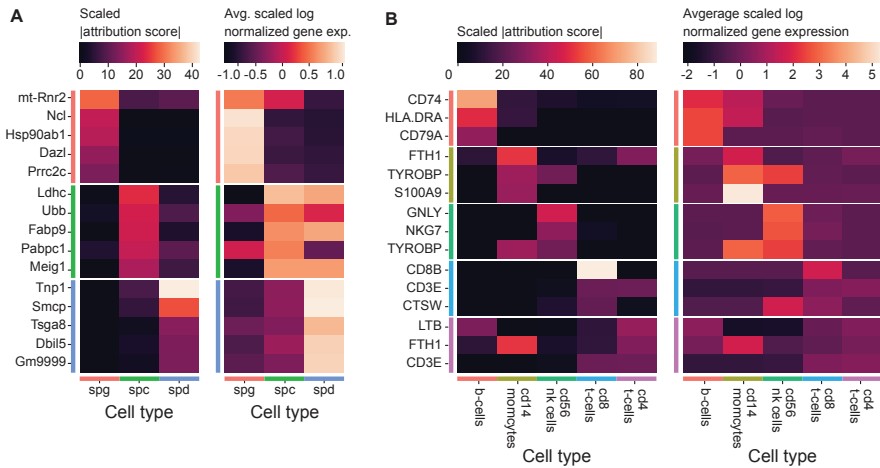

**Figure 2:** Heatmap of DeepLIFT attribution scores after absolute value and scaling by cell type for each predicted cell type and heatmap of average log normalized gene expression scaled by gene for each predicted cell type. a. Top five most important genes for testis data set. b. Top three most important genes for PBMC data set. See Appendix B for extended versions of these plots.

baselines for our model. Max consensus simply chooses the cell type with the most votes. In the event of a tie, this method returns "unknown". We report total accuracy and unconfident cell accuracy for each data set. "Unconfident cell accuracy" refers to the accuracy on only the cells where the underlying tools did not find consensus.

**Simulated Data Sets.** For both simulated data sets, the optimal model has batch size 20, 2 nearest neighbors, and 2 EdgeConv layers. For the simulation with 0.7 de.facScale, 25-dimensional embedding space was optimal, whereas for the simulation with 0.8 de.facScale the optimal value was 40. Table 1 shows our model outperforms all other methods for total accuracy and slightly under performs SingleR for unconfident cell accuracy on the 0.7 de.facScale data.

**Testis Data Set.** Only marker based prediction tools were used for this data set as no labeled reference was easily available. The optimal model for this data set used batch size 20, 2 nearest neighbors, 2 EdgeConv layers, and embedding layer size of 25. Table 1 shows accuracy results, demonstrating our GCN model outperforms all other methods for both total and unconfident cell accuracy, except for the non-parametric approach.

**PBMC Data Set.** The optimal model for this data set used batch size 50, 2 nearest neighbors, 2 EdgeConv layers, and embedding layer size of 25. Table 1 shows accuracy results. For accuracy on unconfident cells, the GCN model places third behind ScType and the non-parametric approach. Our model still outperforms all other methods for overall accuracy.

### 4.2 Feature Interpretation

Figure 2A shows the five most important (as discovered by DeepLIFT) genes by cell type and the expression of these same genes for the testis data set. Interestingly, all of these top genes have uniquely high attribution in their important cell type. The highly attributed genes for a cell type also have relatively high gene expression in that cell type. We also observe high expression of Spermatocyte genes in Spermatid cells. DeepLIFT also indicates genes like Tnp1 that are differentially expressed in those cell types, but not explicitly included as marker genes. Figure 2B shows the three most highly attributed genes by DeepLIFT for each predicted cell type and the scaled gene expression for these genes by cell type for the PBMC data. For B Cells, Monocytes, and NK Cells we see a clear connection between the genes picked out as important by DeepLIFT and the genes expressed by those cell types. However, for CD4 and CD8 T Cells, the expression is not clearly higher for all genes. Importantly, we do observe CD8B as the most important gene for CD8 T Cell classification, a key marker for the cell type. We also observe CD3E (another important marker for all T Cells) as an important gene for both sub types of T Cells. One potential reason for less informative DeepLIFT

scores for CD4 and CD8 T Cells is that the GCN often misclassifies CD8 T Cells as CD4 T Cells. Importantly, the GCN is the only one of our tested methods that can be interpreted using DeepLIFT.

## 5   Discussion

In this work we propose a novel framework for scRNA-seq cell type annotation. Building upon existing annotation tools, we implement an EdgeConv based GCN model to propagate consensus based confident labels to the remaining unlabeled cells. We show an improvement in accuracy over a baseline max consensus algorithm and the average tool accuracy. We also demonstrate the ability to identify important genes for classification via model interpretation with DeepLIFT. The model interpretation is especially valuable for researchers as it has the potential to uncover novel gene markers and provide insight into the model's decisions.

## 6   Acknowledgements

The authors thank Katinka Vigh-Conrad for her help preparing figures and Eisa Mahyari for his helpful discussion. This work was supported by the National Institutes of Health [P50 HD096723-01]. Thanks to the Colorado College Department of Mathematics and Computer Science for supporting this research via the faculty summer research stipend.

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

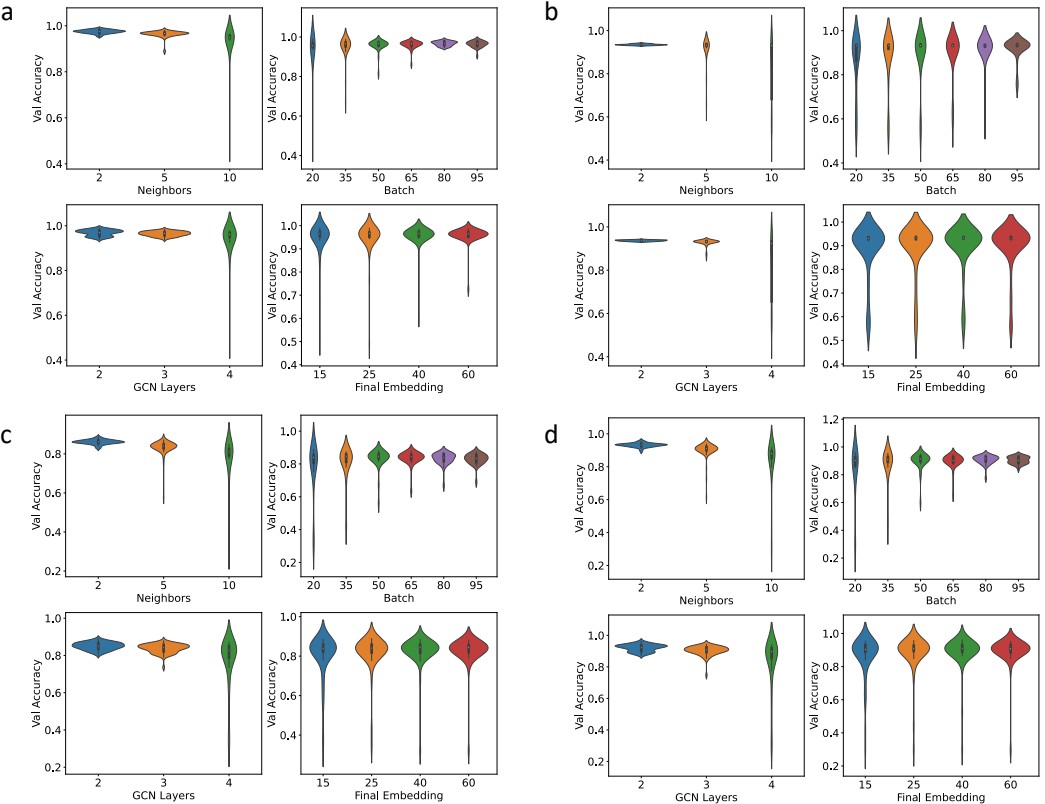

**Figure 3:** Spread of validation accuracy scores as a function of various hyperparameters. The hyperparameters included are number of neighbors, batch size, GCN layers, and final embedding layer size. a. Testis data set. b. PBMC data set. c. Simulation 0.7 data set. d. Simulation 0.8 data set

# A    Hyperparameter Search Details

See Figure 3 for details of hyperparameter search on validation set of each data set.

Our model architecture consists of $l$ EdgeConv layers. Each EdgeConv layer consists of one round of message passing along edges of the graph, followed by a dense neural network model that maps from one layer's embedding space to the next layer's. Each node aggregates information using the sum of its received messages (from neighbors and itself). In all of our model architectures, the first layer takes input embedding size 500 and outputs embedding size 1000. The middle layers accept embedding size 1000 and output embeddings of the same size. The final layer accepts embedding size 1000 and outputs final embedding size $e$. Both hyperparameters number of layers $l$ and final embedding size $e$ are included in the hyperparameter search.

# B    Extended DeepLIFT Plots

See Figure 4 for specific gene expression of each highly important gene for all cell types in testis and PBMC data sets.

# C    Github Link

The code for our pipeline used to generate results in this paper is available at `https://github.com/lewinsohndp/scSHARP`.

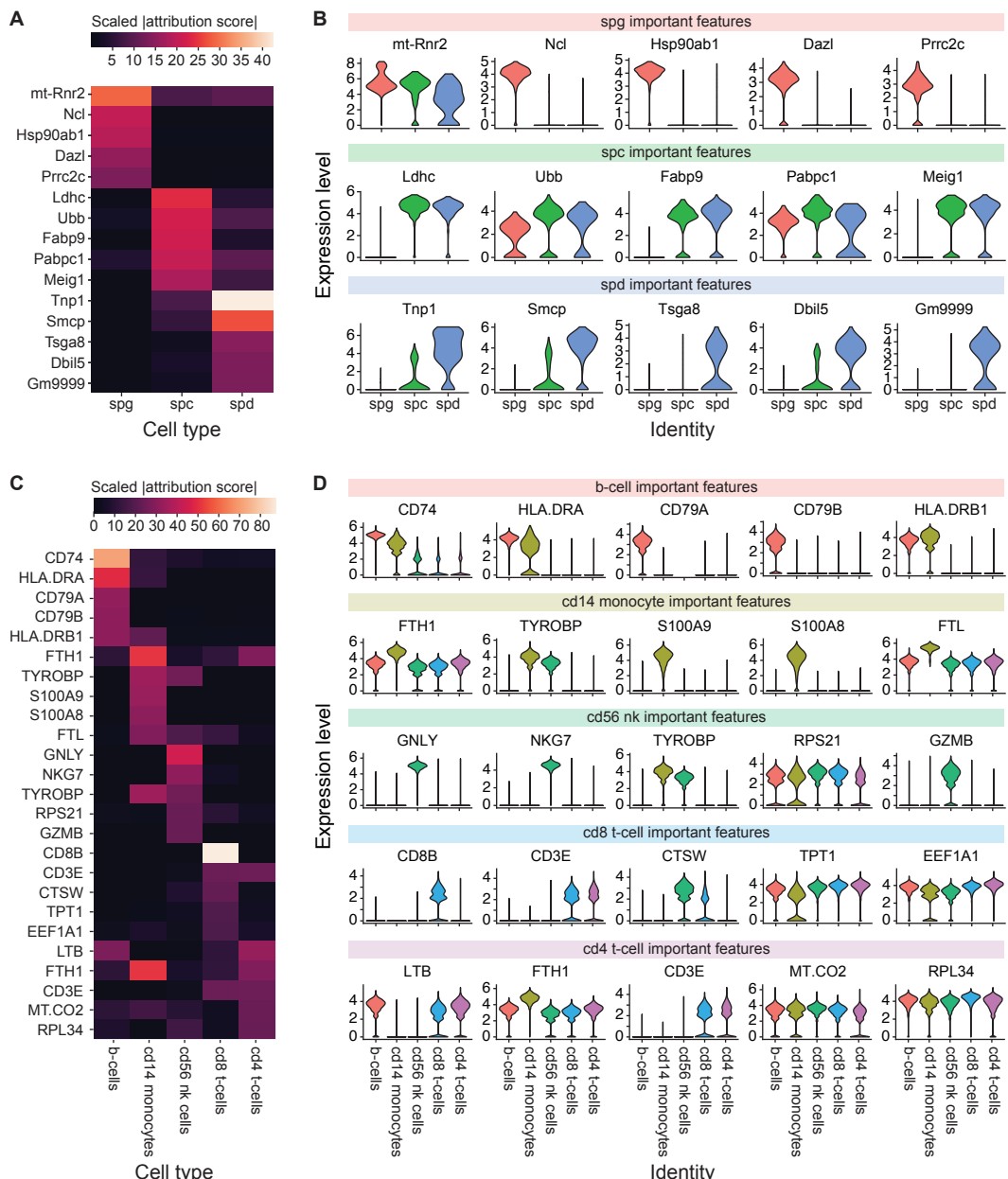

**Figure 4:** Heatmap of DeepLIFT attribution scores after absolute value and scaling by cell type for top five most important features by predicted cell type and violin plot of log normalized expression for each gene. a. Attribution heatmap for testis data set. b. Expression plots for testis data set by predicted cell type. c. Attribution heatmap for PBMC data set. d. Expression plots for PBMC data by predicted cell type.

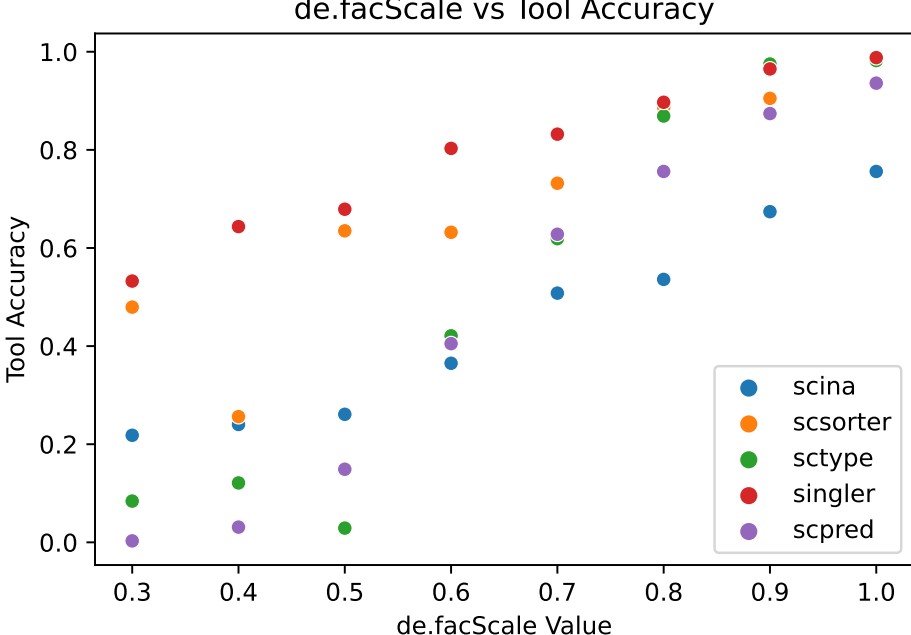

**Figure 5:** Plot showing relationship between de.facScale simulation parameter and component tool accuracy. ScType, SCINA, ScSorter, SingleR, and ScPred were included as component tools.

## D   Discussion of Methods

ScType employs a clustering-based approach that inputs the scRNA-seq cells by genes matrix along with cell type gene markers, and outputs predictions [6]. ScSorter is another clustering-based approach that inputs the scRNA-seq cells by genes matrix along with cell type gene markers. This method recognizes that over-expression of certain marker genes is not present in populations of many cell types and attempts to address this problem [7]. SCINA is another state-of-the-art approach that inputs the same information as ScType and ScSorter, with the added benefit of being much faster. SCINA uses an expectation-maximization algorithm to assign labels [5]. SingleR requires both the input scRNA-seq cells by genes matrix and a labeled reference cells by genes matrix. This method uses correlation between the reference and query sets to extend labels [8]. ScPred requires the same information as SingleR. This method uses feature space reduction to pull out important cell type features and then a machine learning probability-based prediction algorithm [9].

## E   de.facScale Simulation Parameter

See Figure 5 for details on how the de.facScale parameter affects classification difficulty. With de.facScale $\leq 0.5$, the generated data is too difficult for any of the component tools to analyze - all of the component tools do poorly. On the other hand, de.facScale $\geq 0.9$ is too easy - the cell types are well-separated enough in gene space that all component methods are able to classify them correctly. We generated simulated data with de.facScale values of 0.7 and 0.8, as these values produce a challenging, but still attainable, benchmark for classification.

## F   GCN Confusion Matrices

See Figure 6 for confusion matrices from GCN predictions on all data sets. We note that nearly all of these confusion matrices are characterized by a single cell type being the majority of unconfident cells. It is unclear why this is the case for the PBMC and testis data. In the synthetic data, one possible explanation is the way we choose marker genes. We randomly select five of the top ten differentially expressed genes in the simulated data of each cell type as markers (this is standard

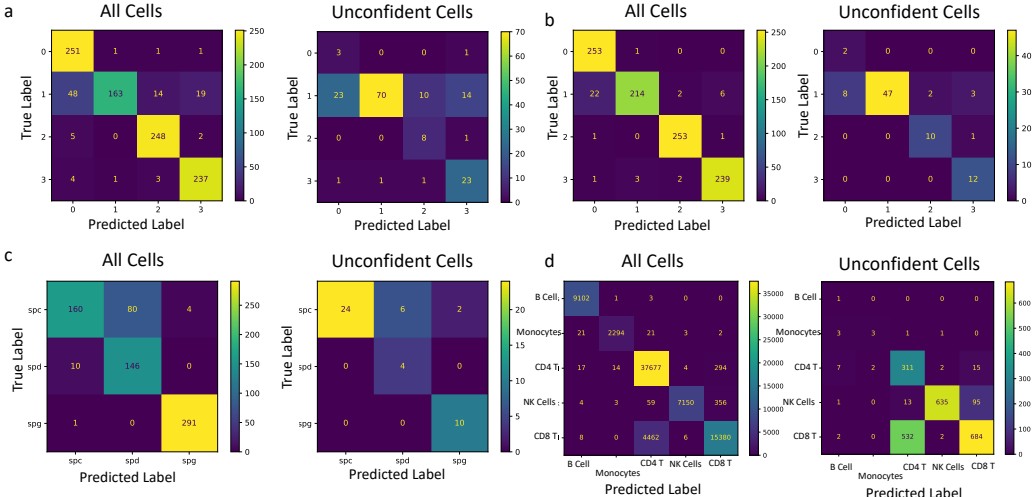

**Figure 6:** Confusion matrices for GCN predictions in all and unconfidently labeled cells. a. Simulation 0.7 b. Simluation 0.8 c. Testis d. PBMC

**Table 2:** Accuracy percent scores for all data sets for both all cells and "unconfident cells": cells for which the underlying methods did not have consensus.

| Method | Simulation 0.7 | | Simulation 0.8 | | Testis | | PBMC | |
|---|---|---|---|---|---|---|---|---|
| | **All** | **Unconf.** | **All** | **Unconf.** | **All** | **Unconf.** | **All** | **Unconf.** |
| Table 1 Best | **90.0 ± .61** | **80.1** | **96.1 ± .22** | **83.3 ± 2.6** | 86.2 ± .12 | 80.9 ± 1.8 | **93.1 ± .05** | **81.8** |
| Non-Parametric | 30.6 | 16.7 | 37.4 | 15.3 | **87.4** | **84.8** | 87.9 | 70.8 |

practice for using Splatter). It is possible this method results in some cell types with better markers than others. This was intentional, as in real-world data not all cell types will always have the same strength of cell type marker. For the actual data sets, this is likely because certain cell types (such as CD4 and CD8 T Cells) are more transcriptionally similar and likely to be misclassified. A common theme in both of these cases is that within each data set, some cell types are inherently easier to classify than others.

# G   Non-parametric Neighbor Majority Label Propagation

We implemented a non-parametric neighbor majority approach as an additional baseline to test our GCN model. This method operates on the 500D vectors produced as the principal components of the gene expression matrices for each data set. We use similarity in this vector space to propagate labels from confident nodes to the remainder of the population. This is similar to the message passing step in our GCN model, with the difference that this method does not use a neural network to encode/decode messages. Each round of message passing, each node's label is updated as the majority label of its k nearest neighbors (only considering those neighbors who have been labeled thus far). We test three strategies:

- one round of label propagation;
- iterating until less than 5 percent of labels change between epochs; and
- iterating until all cells are labeled or 50 epochs have gone by.

For this experiment, we updated cells in batches of 1000, as constructing full k-NN graphs for our PBMC data set proved computationally intractable. It is important to note batch size 1000 leaves both the simulated and testis data sets fully intact without batching.

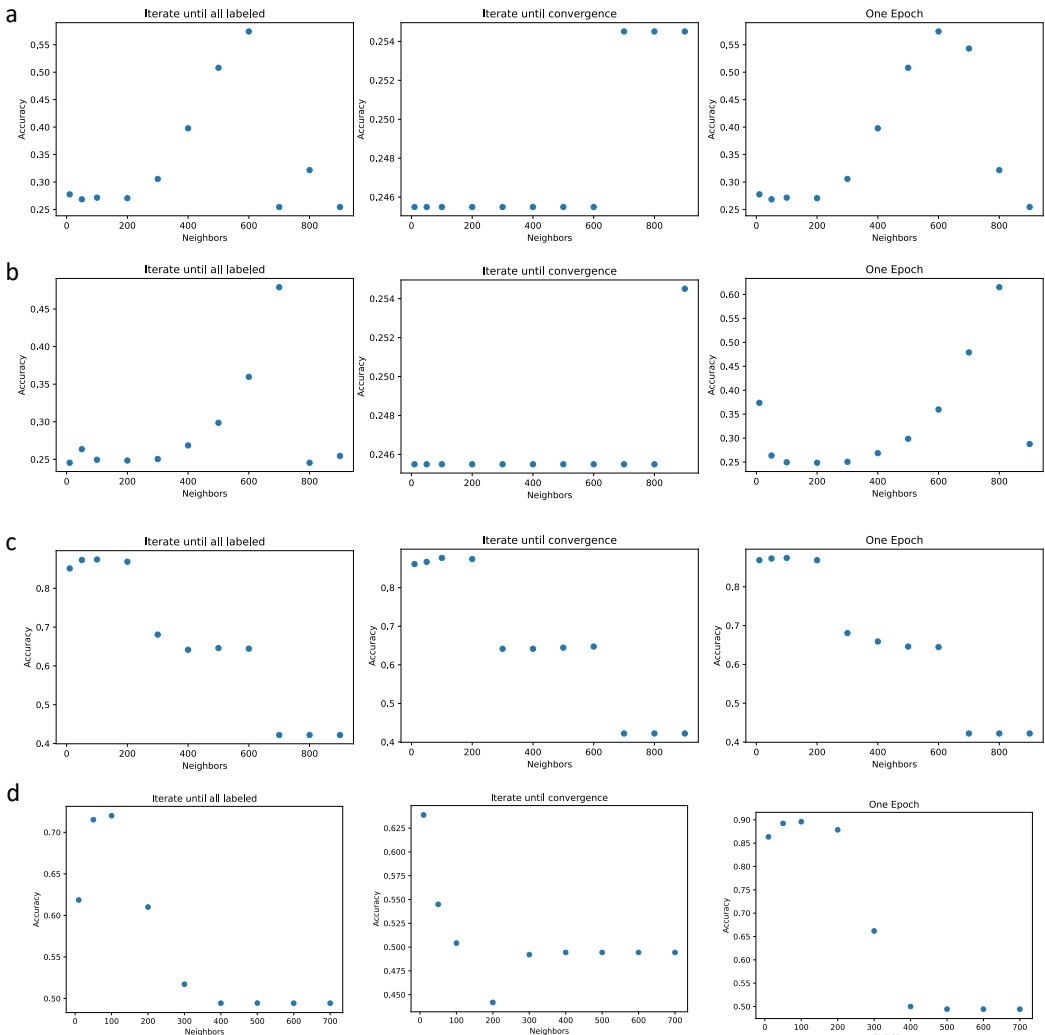

**Figure 7:** Number for neighbors vs accuracy percent for different convergence methods and data sets.
a. Simulation 0.7 b. Simulation 0.8 c. Testis d. PBMC

The results for this method across a variety of k values are shown in Figure 7. We then compared this approach to the best methods from Table 1 in Table 2. To select k, we did a grid search of the same k values and convergence approaches used in Figure 7 and selected the optimal configuration based on a held-out validation set of 20 percent of the confidently labeled cells. For the PBMC data set, only five percent of the data set was used for this hyperparameter search. For the Simulation 0.7, Simulation 0.8, and PBMC data set, running for one epoch with 300, 10, and 200 neighbors respectively was optimal. For the testis data set, running until convergence with 200 neighbors was optimal. These results show the non-parametric approach far under performs our method in the simulated data sets. However, this approach slightly out performs our GCN method in the testis data set. Additionally, it slightly out performs our GCN method in the PBMC unconfidently labeled cells. Although this non-parametric neighbor majority approach does slightly out perform ours in the testis data set and in the PBMC unconfident cells, this method is not differentiable and so does not allow for gene-level model interpretation via DeepLIFT, as our method does. Additionally this method of label propagation is not guaranteed to label all of the cells in the data set - for PBMC, the best performing variant of this method left 31 cells unlabeled.

