# OpenReview forum: "Consensus Label Propagation with Graph Convolutional Networks for Single-Cell RNA Sequencing Cell Type Annotation"
_logconference.io/LOG/2022/Conference — LoG 2022 Poster_

### Official Review · Reviewer_TzwA · 2022-10-02

**Overall Score:** 6
**Confidence:** 4

**Review:**

Summary: the paper proposes a GNN model to extract a consensus cell type labelling from an ensemble of labelling tools. It first constructs a kNN cell graph which connect cells based on their rna-seq profiles. Using a majority rule, each cell is either "confidently" or "unconfidently" assigned a cell type. The former are used as training/validation sets, while the latter is solely used for evaluation.

Comments:
- There is a lack of discussion around existing art. We suggest the authors add a separate section in the appendix that 1) gives some more technical details about the existing cell type labelling method classes, 2) describes the five chosen algorithms in details, and 3) discusses existing consensus label propagation approaches. There are a few that have been introduced in the past and it would make a stronger case for the proposed approach if it were benchmark against these.
- The method description is somewhat lacking details around 1) graph creation and 2) node features. While the reader can perhaps infer it from experience, it would be useful to have it clearly stated. For instance, it is not clear if PCA features are used to generate the graph, to featurize the nodes, or both.
- The sentence "Batches are randomly selected and our graph is reconstructed for each batch and epoch." is a bit confusing and would benefit from some elaboration.
- (minor) The cell count normalisation equation on page 2 should be properly formatted.
- The choice of 0.7/0.8 de.facScale for synthetic datasets seems arbitrary and is not well motivated. Does it relate to real-world datasets? It might also be interesting to show the evolution of metrics on a larger range of de.facScale, potentially shedding light on some sort of transition. This might help identify where taking a consensus approach would be especially beneficient.
- The paper woud benefit from more discussion and analysis around the "unconfident" cell sets, particularly in terms of cell type composition.
- Considering the imbalance in cell types counts in some datasets (and perhaps in unconfident sets), accuracy seems like a potentially misleading metric. Using confusion matrices might also provide some insights in terms of which cell types are more susceptible to incorrect labelling.
- (minor) It might be interesting to investigate the effect of overfit to confident labels on performances. Similarly, showing the proportion of confident labels that were corrected, rightly or wrongly, by the algorithm.
- The DeepLift interpretation appear circular in reasoning as it seems that most, if not all, of the 5 base algorithms chosen rely on known marker genes for each cell types. It then comes at little surprise that the proposed algorithm conserves this information.
- (minor) DeepLift does not explicitly makes use of the graph structure. Have the authors looked into some of the multiple GNN interpretation tools?

---

### Official Review · Reviewer_oLZG · 2022-10-03

**Overall Score:** 8
**Confidence:** 4

**Review:**

This paper presents a graph representation learning methodology to annotate cell types from single-cell gene expression. The proposed method integrates the predictions from multiple cell-type annotation approaches through a GCN that propagates confidently-labelled cell-type labels, with consistent gains in prediction performance.

Strengths:
* This work leverages graph neural networks to address a central research problem in bioinformatics — cell-type annotation from single-cell transcriptomics. The applications in this area are broad and relevant to the scientific community.
* The proposed approach integrates the predictions of several popular cell-type annotation methods. The GCN method is simple and consistently outperforms the baselines.
* The experiments are well-designed and fairly comprehensive (7 baselines, 4 datasets, confident vs unconfident) given the limited space.
* The authors use DeepLIFT (a feature attribution method) to interpret the model’s predictions, with potential to reveal novel gene markers. The feature interpretation insights in the results section are valuable.
* Overall, this paper is well-written and easy to follow.

Weaknesses:
* This is not a weakness per se, but rather a question that I would like to see addressed in the paper: how does the GCN approach compare to a non-parametric “neighbour-majority” baseline that uses the same k-NN graph? In this baseline, every cell would be labelled as the most common cell type among its neighbours. One can iterate until a) convergence or b) the whole graph is labelled. Labels can either be updated once only or multiple times (i.e. whenever the neighbour majority label changes). This experiment would be interesting to understand the behaviour of the GCN and study the influence of graph subsampling (i.e. one can use the full cell-cell graph as this baseline is non-parametric).
* Error bars are not included — how robust are the results across multiple runs? What are the standard errors? Is the improvement over e.g. the “max consensus” baseline statistically significant?

Minor comments and questions:
* What is the difference between reference and query cells? My understanding is that cell-type labels are assumed to be known for the reference dataset. It would be good to highlight the definition of these two cell categories in the main text (as well as their implications for downstream analysis). It would also be helpful to explicitly explain the difference between a) confident vs unconfident cells and b) reference vs query cells.
* Figure 1 suggests that the model uses cell-type markers when these are available — how is this information incorporated into the architecture, exactly?
* What is the homophily of the cell networks for different numbers of nearest neighbours k? Does the network homophily correlate with prediction accuracy?
* Typo: “FACs” -> “FACS”

In the light of the simplicity of the proposed method, sound experimental design, and satisfactory integration of multiple cell-type annotation methods, I recommend the acceptance of this paper.

---

### Official Review · Reviewer_fK1t · 2022-10-21

**Overall Score:** 5
**Confidence:** 4

**Review:**

Summary:

The authors present a Graph Convolutional Network (GCN) based approach to automate cell type annotation for single-cell RNA sequencing data. They build on multiple state-of-the-art labeling approaches to come to a consensus on highly confident cells. Then they implement a semi-supervised GCN model to propagate labels from confidently labeled cells to unconfidently labeled cells. They also use DeepLIFT for feature interpretation to determine the attribution score of each gene on a  given cell type.

Comments:

The overall readability of the manuscript is good
Even though the authors compare with a few state-of-the-art methods, it would improve the manuscript if there are comparisons with state-of-the-art Graph-based approaches such as scDeepSort or any other deep learning-based approach. This would also enable the comparison of DeepLIFT analysis on other differentiable models (line no. 60).
For pbmc analysis, the rationale behind combining the cell types to reduce the number of classes from 10 to 5 is unclear as these final 5 cell types are the most feature-rich and presumably easy to classify. It is also unclear what it means to remove the types not purely sorted by FACs. (line 86 - 92). Considering that the ‘ScType’ [6] paper (one of the approaches used for comparison) includes all the 10 cell types, is there a particular reason for this reduction?
Table 1 :
It is unclear what it means by unconfidently labeled cell accuracy for Max consensus approach. Line 103: “The unconfident cell accuracy refers to cells that had no majority vote (max consensus) for a cell type by underlying tools.”
It would be good to compare the misclassification rate / accuracy of each cell type for all approaches along with overall cell type accuracy.
Figure 2:
The DeepLIFT attribution scores for CD4 and CD8 T cells are curious. The authors do mention this could be due to the low classification accuracy of GCN on these cells. However, considering the fact that these cells can be quite distinctly separated during clustering analysis of pbmc data, this behavior of GCN on these cells is worth exploring and might need a bit more explanation. It would be good to see the classification accuracy of just these cells on all the methods compared.

The idea of using GCN to propagate consensus-based confident labels to the remaining unlabelled cells is intriguing. However, the manuscript would highly benefit if there are a few more comparisons and analyses as mentioned.

---

### Official Review · Reviewer_JAtC · 2022-10-22

**Overall Score:** 6
**Confidence:** 4

**Review:**

This work proposes a simple but effective strategy based on a graph neural network to improve cell type annotation.

Strengths:
- The proposed idea is reasonable and has useful applications. Ground truth annotations are usually unavailable for real/novel scRNA-seq datasets, and often scientists rely on multiple tools, with simple consensus metrics such as mean/max. The proposed method addresses the need for a better, data-driven consensus strategy.
- The selection of reported baselines is sufficient to draw the conclusions.
- The datasets used for the experiments are diverse, including both simulated and public data.

Weaknesses:
- The novelty of the method is somehow limited. The method combines existing approaches in the single-cell and GNN domains. However, the resulting application is novel and potentially useful.
- The authors should add confidence intervals in Table 1.
- The authors should add more details about data preprocessing, in particular, graph preparation. How does the choice of the number of cell neighbors affect the results? And the robustness? That seems a critical and arbitrary component of the model, as it critically affects message passing.

Overall, I think the paper is well-written and clear, and presents a valuable application with convincing results. Authors should focus on improving novelty and better evaluating design choices.

---

### Meta-Review · Area_Chair_GWvv · 2022-11-11

**Confidence:** 4
**Recommendation:** Accept

**Meta Review:**

This paper combines robust methods from the graph neural network literature to solve the very important problem of annotating cell types in transcriptomics. More specifically, the methods pools the results of various SOTA cell types annotation methods to identify strong components that are then spread via a GCN approach.

Reviewers broadly agree on the merits of the paper. It tackles a very important problem, that of automatically annotating cell type in transcriptomics, which has broad applications. The experiments are convincing, with good comparisons to baselines and using diverse datasets. Some reviewers pointed at limited information of the data preparation, which I believe has been correctly answered making the results reproducible, and to limited novelty, the paper using several existing techniques. The combination of techniques and use of the GCN consensus is however novel enough.

Conclusion and recommendation: the significance for applications and clear validation outweigh concerns on limited novelty and I therefore recommend to accept this paper.

---

### Decision · Program_Chairs · 2022-11-23

Accept (Poster)